# The piston Riemann problem in a photon superfluid

Abdelkrim Bendahmane[1], Gang Xu [1], Matteo Conforti [1✉], Alexandre Kudlinski[1], Arnaud Mussot [1] & Stefano Trillo [2✉]

Light flow in nonlinear media can exhibit quantum hydrodynamical features which are profoundly different from those of classical fluids. Here, we show that a rather extreme regime of quantum hydrodynamics can be accessed by exploring the piston problem (a paradigm in gas dynamics) for light, and its generalization, named after the celebrated mathematician Riemann, where the piston acts on a concomitant abrupt change of photon density. Our experiment reveals regimes featuring optical rarefaction (retracting piston) or shock (pushing piston) wave pairs, and most importantly the transition to a peculiar type of flow, occurring above a precise critical piston velocity, where the light shocks are smoothly interconnected by a large contrast, periodic, fully nonlinear wave. The transition to such extreme hydrodynamic state is generic for superfluids, but to date remained elusive to any other quantum fluid system. Our full-fiber setup used to observe this phenomenon in temporal domain proves to be a versatile alternative to other platforms currently employed to investigate the hydrodynamical properties of quantum fluids of light.

[1] CNRS, UMR 8523—PhLAM—Physique des Lasers Atomes et Molécules, Univ. Lille, Lille, France. [2] Department of Engineering, University of Ferrara, Ferrara, Italy. ✉email: matteo.conforti@univ-lille.fr; stefano.trillo@unife.it

The flow of light can exhibit hydrodynamic-like properties which are characteristic of the collective behavior of quantum many-body systems (e.g., liquid helium or ultracold atomic gases[1]), hence showing properties which are characteristics of superfluids rather than conventional classical fluids (water, gases)[2]. This requires photons to be significantly interacting through medium nonlinearities. To date many fascinating achievements in this area have exploited two main platforms, both based on spatial effects. The first one relies on light confined in a cavity[3], and exploits polariton fluids in semiconductors[2,4–6]. The second one exploits cavityless paraxial propagation, with photon–photon interactions mediated by the repulsive (or defocusing) nonlinearity of the medium[7,8], which have permitted the observation of hydrodynamic phenomena such as dispersive shock waves (DSWs)[9–13] (a hallmark of dispersive hydrodynamics observed in other superfluids[14–16]), Bogoliubov excitations and their interference[17–19], superfluid flow around an obstacle[20], order–disorder topological transitions[21], and blast waves[22]. In this regime, the mathematical framework is the fully conservative nonlinear Schrödinger equation (NLSE), which implies a hydrodynamical description with intensity and phase gradient of the light field playing the role of density and velocity of a standard fluid, in full analogy with Gross-Pitaevskii mean-field description of cold gases[1].

In this paper, we show that a different fiber-based platform, where temporal nonlinear dynamics with normal dispersion is akin to paraxial propagation in defocusing media, can be successfully employed to detect a rather extreme hydrodynamic state of light. This regime originates from the investigation of the photonic analogue of the generalized piston problem in gas dynamics. In particular, we experimentally address the problem of the flow of a photon fluid initially prepared to present a step-like variation in both velocity and density which realizes the canonical problem named after Riemann (i.e., a well-known building block for the mathematical description of any flow[23,24], but never implemented experimentally). The velocity jump mimics all-optically a piston set impulsively into motion, which acts either on a constant density photon fluid (thus realizing the standard piston problem of gas dynamics[25,26]) or in conjunction with a step density change (most general Riemann problem). In both cases, we demonstrate a transition from a regime reminiscent of piston dynamics in standard gases to a regime that shows no similarity whatsoever in standard fluids[27–29] and has never been observed in other quantum fluids[30]. The former regime occurs at relatively low piston velocities and is characterized by the formation of expanding rarefaction wave pairs (retracting piston) or DSW pairs (pushing piston), the naturally dispersive counterpart of classical shock waves in ideal gases. Conversely, above a critical velocity marked by the onset of cavitation in the DSW pairs, the photon fluid is observed to undergo a transition to an extreme state where the shocks become smoothly connected through a fully nonlinear periodic wave of large contrast instead of a constant density state. This regime can be regarded as the manifestation of the fully developed dispersive hydrodynamic character of superfluid-type of flow, and we show its onset to be in quantitative agreement with predictions from Whitham modulation theory of the NLSE[27–29]. We emphasize that, in spite of a recent growing interest in Riemann problems in different contexts of dispersive hydrodynamics[31–39], to date the experimental observations are limited to simpler discontinuous jumps in the density (see refs. [32,33] in the defocusing case, and also refs. [35–37] for the focusing unstable case). Conversely, observing the wave regimes emerging from the piston problem and its generalization requires to face the twofold challenge of engineering a high degree of control over the input phase gradient of the field, while operating in a truly nonlinear dispersive setting

where even weak dissipation is suppressed. Our experiment, which achieves both goals, proves that fiber-based platforms allow to measure the DSW dynamics with unprecedented precision[32,40–47]. Moreover, they constitute, due to their advantages (ultra-long propagation lengths, fine tuning of initial conditions and fiber parameters, loss compensation to achieve truly conservative dynamics), a mature alternative to paraxial fluids of light to detect the full richness of superfluid-like transitions, which are otherwise challenging to observe in other quantum fluids.

## Results

**Overview of the piston problem.** In the framework of gas dynamics and classical shock waves (CSWs) theory[23,48–52], the piston problem has the canonical solution (see ref. [25,26]) schematically illustrated in Fig. 1a, d. When the piston compresses the gas at rest on its right (Fig. 1a), a CSW emerges that travels ahead of the piston with supersonic velocity (dictated by so-called Rankine-Hugoniot condition[53], see Supplementary Information Note 1), whereas a retracting piston (Fig. 1d) produces a smooth rarefaction wave (RW). Our aim is to implement the analog problem for a photon fluid. To this end, we exploit its conceptual identity with a Riemann problem (see Supplementary Information Note 1 for mathematical details), where the physical piston is replaced by a suitably prepared initial condition characterized by a stepwise variation of the fluid velocity over a constant density. In gases, the latter ideally produces a bi-directional replica of the piston-driven CSW or RW, as sketched in Fig. 1b, e. In the dispersive regime characteristic of the photon fluid, according to modulation theory (or Whitham averaging[27,29,54–58]) for the NLSE, the RWs remain essentially unaltered due to their smoothness (Fig. 1f), whereas the CSWs are turned into expanding DSWs (Fig. 1c). The formation of the two DSWs, however, is expected to exhibit critical behavior[27,29,56]. Indeed above a critical amplitude of the velocity jump where the two DSWs start to cavitate, they become connected by a nonlinear periodic wave instead of a constant background (as it is always the case in gases). This marks a dynamic transition to a regime

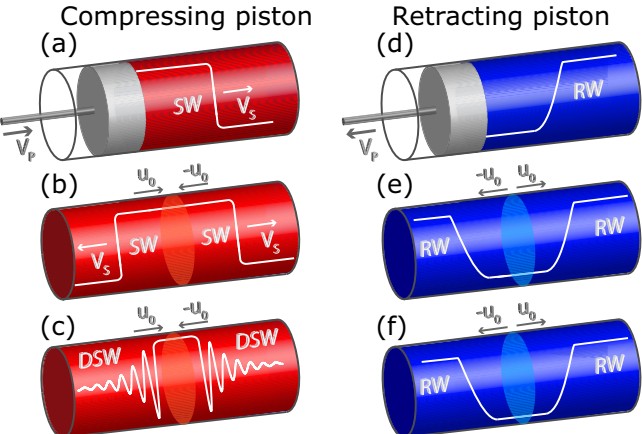

**Fig. 1 Schematic of the physics and typical evolutions of classic vs. dispersive piston problems. a, b, c** Shock wave dynamics for "pushing" piston (white curves show qualitatively the density): (**a**) compressing piston producing a shock wave in an ideal (non-dispersive) fluid; (**b**) shock-shock produced via stepwise initial velocity profile (Riemann problem); (**c**) DSW-DSW via Riemann problem in a dispersive photon fluid; d,e,f: Rarefaction wave dynamics for "retracting" pistons: (**d**) single RW; (**e**) RW-RW from the Riemann problem; (**f**) similar RW-RW in a dispersive photon fluid.

which is unique to the photon fluid. We report here the observation of such a dispersive hydrodynamic transition.

**Theoretical developments from the Riemann problem.** Our experiment is concerned with the implementation of the Riemann problem for the following defocusing NLSE, which governs the propagation along an optical fiber of the electric field envelope $E = E(Z, T)$ that modulates the light at the optical carrier pulsation $\omega_0$[59]

$$i\frac{\partial E}{\partial Z} - \frac{k''}{2}\frac{\partial^2 E}{\partial T^2} + \gamma |E|^2 E = 0, \quad (1)$$

where $Z$ is the propagation distance along the fiber and $T$ stands for the retarded time (in a frame moving at the group velocity $V_g = dk/d\omega|_{\omega_0}^{-1}$, $k = k(\omega)$ being the fiber mode wavenumber[59]). The parameters $k'' = d^2k/d\omega^2|_{\omega_0}$ and $\gamma$ stand for the group-velocity (or second-order) dispersion and the fiber nonlinear coefficient due to the Kerr effect, respectively[59]. Henceforth, we make use of the values that characterize our experiment: $k'' = 170$ ps$^2$/km and $\gamma = 3$ (W km)$^{-1}$. In these units $|E(Z,T)|^2$ gives directly the optical power in Watts.

We operate in the defocusing regime of the NLSE ($\gamma k'' > 0$), where a clear connection to CSWs of gas dynamics (or hydraulic jumps[60]) does exist. It becomes manifest by applying the so-called Madelung transform $E(T, Z) = \sqrt{P_0}\sqrt{\rho(t, z)}\exp(-i\int_{-\infty}^{t} u(t', z)dt')$, which allows to formulate the NLSE in fluid-dynamics form:

$$\rho_z + (\rho u)_t = 0 ; \quad (2)$$

$$u_z + \left(\frac{u^2}{2} + \rho\right)_t = \frac{1}{4}\left[\frac{\rho_{tt}}{\rho} - \frac{(\rho_t)^2}{2\rho^2}\right]_t, \quad (3)$$

where we set $z = Z/Z_0$, $t = T/T_0$ with $Z_0 \equiv (\gamma P_0)^{-1}$ and $T_0 \equiv \sqrt{k''/(\gamma P_0)}$, $P_0$ being a reference power. In the following we fix $P_0$ to be the power of left ($t < 0$) side of the input step. By neglecting the right hand side containing higher-order derivatives arising from dispersion (also known as quantum pressure term[9]), Eqs. 2 and 3 are identical to the dispersionless vector Eulerian conservation law that rules the dynamics of the one-dimensional flow in an isentropic gas with pressure law $p \sim \rho^2$[23]. Here, the normalized power $\rho(t, z) = |E(T, Z)|^2/P_0$ plays the role of local gas density, whereas the normalized instantaneous frequency deviation from the carrier (usually denoted as chirp) $u(t, z) = \Delta\omega(T, Z)$ $T_0$ is equivalent to gas velocity. Here $\Delta\omega(T, Z) = -d\phi/dT$ is the dimensional chirp, expressed in terms of the envelope phase $\phi = Arg[E(Z, T)]$. In this limit, which henceforth will be referred to as the dispersionless NLSE, the resulting system is hyperbolic, thereby admitting weak solutions known as CSWs, which describe traveling jumps in density and velocity.

In particular, the step-like initial condition shown in Fig. 1b, i.e., a decreasing jump in velocity (chirp), produces, in the dispersionless limit, two CSWs propagating in opposite directions as illustrated by the black solid lines in Fig. 2a, b. As shown in Fig. 2, the CSWs feature two fronts that connect either the initial left ($\rho, u_0$) or the right ($\rho, -u_0$) quiescent state, to a plateau or intermediate state of constant density $\rho_i > \rho$ and zero velocity ($u_i = 0$), which emerges spontaneously and turns out to be expanding as the two shocks propagate. The explicit expression of the intermediate density

$$\rho_i = \left(\frac{u_0}{2} + \sqrt{\rho}\right)^2 \quad (4)$$

follows from a simple-wave approach for hyperbolic equations, which allows also to express the velocities of the shocks from the well-known Rankine-Hugoniot condition[23,53]. In terms of the

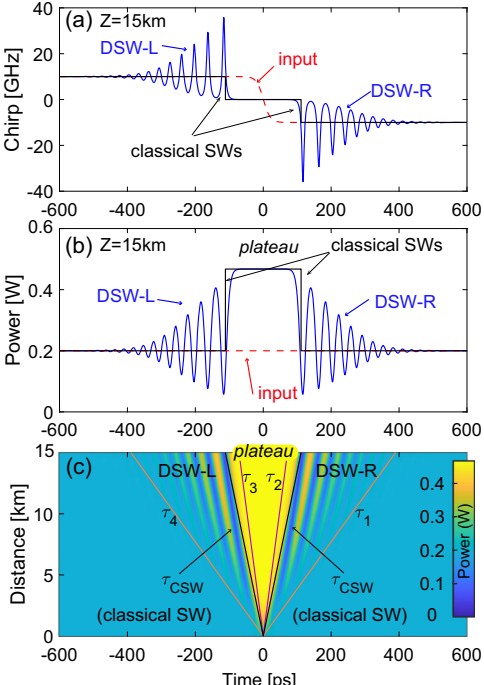

**Fig. 2 Formation of two DSWs ruled by the full NLSE (with parameters of the experiment) contrasted with the CSWs of the dispersionless (isentropic gas-dynamic) case.** Snapshot of chirp $\Delta f(T, Z) = u(t, z)/(2\pi T_0)$ (**a**) and power $|E(Z, T)|^2 = \rho(z, t)P_0$ (**b**) at the output $Z = 15$ km (blue solid line). The input (dashed red) is a frequency step $\Delta f(z = 0) = \pm 10$ GHz (see also Methods) on a constant power $P_0 = 200$ mW. **c** Evolution of power in $T - Z$ plane showing the expansion fans of the two left (DSW-L) and right (DSW-R) shocks delimited by edge velocities in Eq. 6; The CSWs are the solid black lines in (**a**, **b**), while their velocities (Eq. 5) are given by the oblique black lines in (**c**).

self-similar variable $\tau = t/z$, such velocities reads

$$\tau_{CSW}^{\pm} = \mp\left(\frac{u_0}{4} - \sqrt{\rho}\right). \quad (5)$$

In Fig. 2a–c, we also contrast this gas dynamics scenario with the corresponding dispersive dynamics obtained by numerical integration of the full NLSE. As shown, in the latter case, the two shocks become indeed dispersive, being characterized by expanding fans (see Fig. 2c) where fast oscillations spontaneously appear, connecting the upper and lower quiescent states. The wavetrains that constitute the left (DSW-L) and right (DSW-R) dispersive shocks reflect their nature of periodic nonlinear waves (dn-oidal waves) with strongly modulated parameters[29]. The modulated wavetrain in each DSW spans a temporal interval that ranges from a soliton edge (the inner deep end of the wavetrain) to a harmonic edge (where oscillations become shallower, i.e., quasi linear). These two edges travel with different velocities, say $\tau_{1,2}$ for the DSW-R, which can be predicted by means of Whitham modulation theory (details in Supplementary Information Note 2) and reads as:

$$\tau_1 = \frac{u_0^2 + 3u_0\sqrt{\rho} + \rho}{\sqrt{\rho} + u_0}; \quad \tau_2 = \sqrt{\rho} - \frac{u_0}{2}. \quad (6)$$

It is also clear from Fig. 2c that the edge velocities of the DSW-L are $\tau_4 = -\tau_1$ and $\tau_3 = -\tau_2$ due to symmetry, whereas the CSW are slightly faster than the DSW soliton edges.

The case illustrated above is actually a particular case of the most general Riemann problem such that the initial condition is

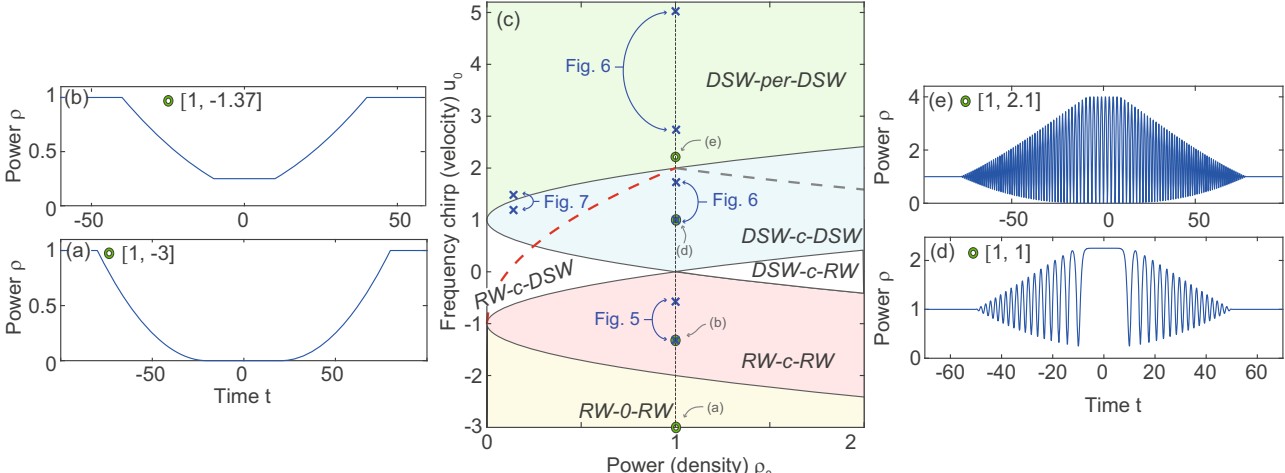

**Fig. 3 The Riemann problem of the NLSE. c** Dynamic transition diagram: domains in the plane of input parameters $(\rho_0, u_0)$ of the step, giving rise to different wave-pair compositions of rarefaction (RW) and dispersive shock (DSW). The two waves in the pair can be connected through a constant state (-c-), zero density (vacuum state -0-); or an unmodulated nonlinear periodic wave (-per-). Above the red (gray) dashed lines, a vacuum is predicted to appear in the DSW-R (DSW-L). **a**, **b** and **d**, **e** are typical examples of output (normalized length $z_L = 20$) from Whitham modulation theory marked with green circles in (**c**). Experimental results are marked with blue crosses. See also Supplementary Information for a video illustrating the transitions between all the cases along the ($\rho_0 = 1$) vertical line and for an extended version of this figure where, for completeness, we also report former experimental results from ref. [32].

step-like in both chirp and power. Without loss of generality (see Methods), we can assume initial conditions characterized by only two parameters $(\rho_0, u_0)$ which describe a left ($t < 0$, $u_L, \rho_L$) to right ($t > 0$, $u_R, \rho_R$) symmetric jump in velocity, from $u_L = u_0$ to $u_R = -u_0$, accompanied by a power jump from $\rho = \rho_L = 1$ to $\rho = \rho_R = \rho_0$. The decay of the step-initial data ruled by the NLSE generates a pair of fundamental waves, each being of the shock or rarefaction type, separated by a constant state or periodic state (see refs. [27,56] and Supplementary Information Note 2). Five different regimes are possible, which are highlighted by domains of homogeneous color in Fig. 3c. When the parameters of the stepwise input $(\rho_0, u_0)$ are changed in such a way to cross the border between domains of different colors in Fig. 3c, a qualitative change occurs in the dynamical appearance of the two wave constituents. In the following, we refer to this change as a dynamic phase transition, due to its reminiscence of the changes of state driven by conventional (e.g., thermodynamic) phase transitions.

Let us briefly examine the different domains in Fig. 3c. The white regions in Fig. 3c correspond to decay into a DSW and a RW connected by a constant power, which we indicate as DSW-c-RW. Experimental evidence for such scenario has been recently reported in fiber optics[32] and spin waves[33], for initial data lying on the horizontal line $u_0 = 0$. Arguably, this is the most common situation which leads to the formation of a shock wave, which is also known as dam breaking problem in hydrodynamics[61].

In this paper, however, we are rather interested in the piston problem, that is a jump in velocity over a homogeneous density, which is described in Fig. 3c by the vertical dotted line $\rho_0 = 1$. Typical theoretical examples (green circles) are shown in Fig. 3a, b, d, e. Let us consider such line, starting from negative values $u_0 < 0$, which is equivalent to have two retracting pistons as sketched in Fig. 1e. As shown by the snapshots depicted in Fig. 3a, b, the two RWs smoothly connect the quiescent input state $\rho = \rho_0 = 1$ to a "rarefied" state of lesser power $\rho_i = (1 - |u_0|/2)^2$ and zero velocity $u_i = 0$, consistently with Eq. 4. The expansion of the right RW is determined by the velocity $\tau_1$ of the faster edge connecting the RW to the input higher density state $\rho = 1$ and the velocity $\tau_2$ of the slower edge where the RW connects to the

rarefied state $\rho_i$. We obtain for these velocities (for $-2 < u_0 < 0$)

$$\tau_1 = 1 + |u_0|; \quad \tau_2 = 1 - \frac{|u_0|}{2}, \tag{7}$$

whereas the left RW expands with symmetric velocities $\tau_4 = -\tau_1$ and $\tau_3 = -\tau_2$. Furthermore, when the two retracting pistons are fast enough, i.e., for $u_0 < -2$, the intermediate state becomes a zero density state or vacuum (see Fig. 3a) and the velocity of the slower edge in Eq. (7) becomes $\tau_2 = |u_0| - 2$.

When $|u_0|$ decreases the RWs become progressively shallower up to the limit $u_0 \to 0$ for which $\rho_i \to 1$ and the density remains obviously flat upon evolution. However, when $u_0$ crosses the line $u_0 = 0$ becoming positive, the behavior drastically changes because now the virtual pistons are pushing the fluid as in Fig. 1b, c. As a result, two DSWs appear, being separated by a flat plateau with density $\rho_i$ which becomes, in this case, larger than the quiescent density $\rho = 1$. This is the case shown in dimensionless form in the example in Fig. 3d and discussed above in more detail with reference to Fig. 2a–c. At the critical value (threshold) $u_0^{th} = 2$, the amplitude of the oscillations in the DSW reach its maximum and the DSWs start to cavitate (the bottom of the oscillations touches zero power), while the plateau shrinks to zero (i.e., the DSWs are glued back to back). Increasing $u_0$ further above this threshold, a phase transition to a peculiar regime occurs. This pattern features two DSWs connected by an unmodulated nonlinear periodic wave (DSW-per-DSW), as displayed in Fig. 3e (more theoretical details are given in Supplementary Information Note 2). This regime, which was firstly pointed out by Bikbaev[56], is a hallmark of the dispersive hydrodynamic behavior of the photon fluid, which bears absolutely no similarity in gas dynamics. Importantly the transition from a DSW-c-DSW to a DSW-per-DSW (cyan to green domain in Fig. 3c) is not a prerogative of the vertical line $\rho_0 = 1$, which describes the canonical piston problem. For generic Riemann step-initial data the threshold reads as (for $\rho_0 < 1$)

$$u_0^{th} = 1 + \sqrt{\rho_0}, \tag{8}$$

which correctly reduces to $u_0^{th} = 2$ for the pure piston problem ($\rho_0 \to 1$).

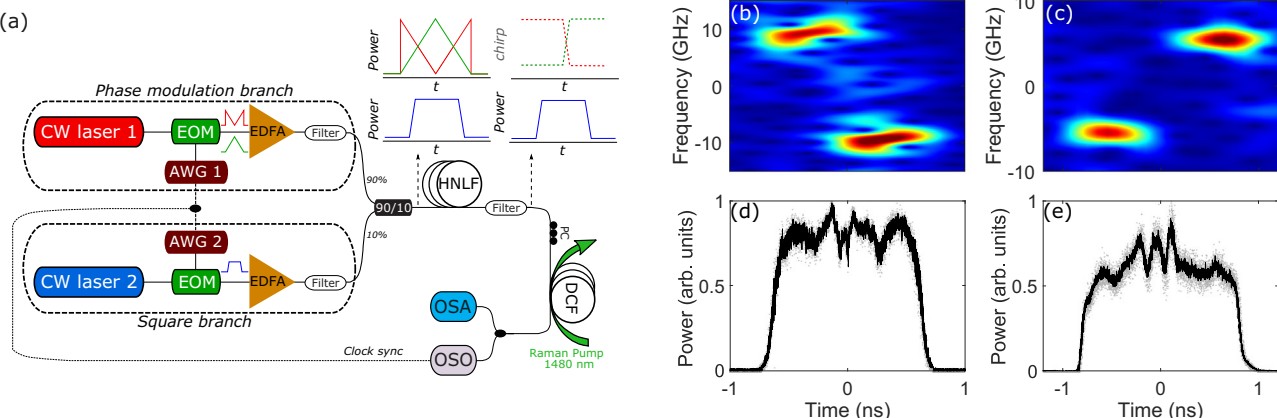

**Fig. 4 Sketch of the experimental setup and characterization of the initial condition. a** EOM electro-optic modulator, EDFA erbium-doped fiber amplifier, AWG Arbitrary wave generator, OSO optical sampling oscilloscope, OSA Optical Spectrum Analyser, HNLF highly nonlinear fiber, $L_H = 500$ m, DCF dispersion compensating fiber, $L = 15$ km, PC polarization controller. The choice of intensity modulation (M- or triangularly-shaped sketched in red and green, respectively) in the upper block (Phase modulation branch) is transformed into a stepwise chirp (either descending or ascending) of the beam at $\lambda_2$, which is also intensity modulated to produce square pulses in the bottom block named Square branch, to compose the input of the DCF. **b–e** Experimental characterization of the input to the DCF: (**b, c**) spectrograms $\Delta f$ vs. time $T$ (dark red and blue stands for the maximum and minimum power density, respectively); (**d, e**) Power profile of the frequency modulated rectangular pulse. Left column (**b, d**) and right column (**c, e**) refer to the case of descending and ascending step-like frequency variation mimicking a pushing or retracting pair of pistons, respectively.

**Experimental results**. In order to observe the physics described above, we designed a full-fiber setup which exploits state of the art telecommunication technology as sketched in Fig. 4a. The main idea behind the preparation of the Riemann-like input is to use two laser sources at slightly different wavelengths suitably modulated in amplitude, which are combined and injected in a first highly nonlinear fiber (HNLF). The HNLF has the key role of transforming the amplitude modulation (M- or triangular shape) of one of the laser sources into the frequency modulation of the other source via cross-phase modulation (XPM). A detailed description of the experimental apparatus can be found in the Methods section.

In Fig. 4b–e we display the results of the experimental characterization of the input, i.e., the Riemann initial condition, before they are launched into the main fiber, namely a dispersion compensating fiber (DCF), which features normal dispersion $k'' > 0$, so to operate in the defocusing regime. Futhermore, linear loss are almost perfectly compensated with a counter-propagative Raman pump in the DCF which allows to investigate the dynamics predicted by the integrable NLSE. The case of descending step-like variation produced via the M-shaped modulation is shown Fig. 4b, d (pushing piston in Fig. 1). In particular, Fig. 4b shows the measured spectrogram, i.e., the frequency deviation $\Delta f = \Delta\omega/2\pi$ from the input carrier frequency as a function of time $T$ (see Supplementary Information Note 4 for further details). The pulse clearly exhibits an abrupt jump in frequency around $T = 0$, with chirp excursion between $\pm\Delta f_0$ with $\Delta f_0 = 9.4$ GHz, and an estimated rise time (10 to 90%) $T_r = 50$ ps.

Being symmetric around $\Delta f = 0$, this frequency modulation realizes the step-like variation from $u_0$ to $-u_0$ formerly introduced in Figs. 1, 3, with $u_0 = 2\pi\Delta f_0\sqrt{k''/(\gamma P_0)}$.

The same type of measurements performed for the choice of triangular driving signal is reported in Fig. 4c, e (pulling piston in Fig. 1). The spectrogram in Fig. 4c clearly shows the ascending nature of the frequency jump. In this case, however, in order to have tolerable distortion of the plateau of the square pulses, the achievable frequency jump is limited to a lower excursion $\Delta f_0 = 5.4$ GHz. The outcome of the experiment is summarized in Figs. 5–7. Importantly, in all regimes, we find that the most stable and repeatable configuration is to operate at the fixed maximum

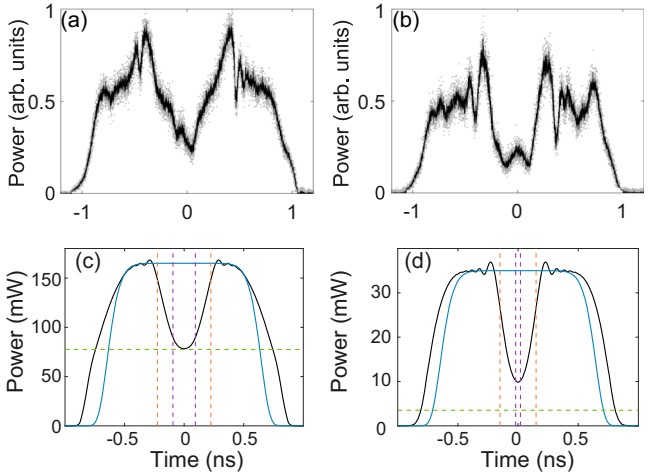

**Fig. 5 Retracting piston experiment.** Rarefaction wave pair produced by an increasing input jump in frequency with $\pm\Delta f_0 = 5.4$ GHz, and power: (**a**) $P_0 = 165$ mW ($u_0 = -0.63$); (**b**) $P_0 = 35$ mW ($u_0 = -1.37$). **c, d** Simulations based on the NLSE. The dashed vertical lines stand for edge hydrodynamic velocities calculated from Eq. 7. The horizontal green line stands for $\rho_i P_0$, with $\rho_i$ from Eq. 4. The blue curves stand for the input super-Gaussian pulse profile on which we superimpose the stepwise chirp jump (see Methods for details).

achievable $\Delta f_0$ (i.e., the values illustrated in Fig. 4b, c or Fig. 7a) and tune the effective velocity of the piston $u_0$ by changing the power of the modulated square pulse in input to the DCF, recalling that $u_0$ scales like $u_0 \propto \Delta f_0/\sqrt{P_0}$. Let us start with the case of negative $u_0$, which corresponds to the retracting piston producing the RW-c-RW case (pink area in Fig. 3c). The output power profiles in Fig. 5a, b clearly show the formation in the center of the nearly square pulse of a wide hole or dark region.

The smooth edges of this hole constitute two optical RWs, that are driven by the initial condition that acts like a pair of retracting pistons. In Fig. 5c, d the corresponding output profiles obtained from numerical integration of the NLSE (1) show a good qualitative agreement with the experiment. The experimental

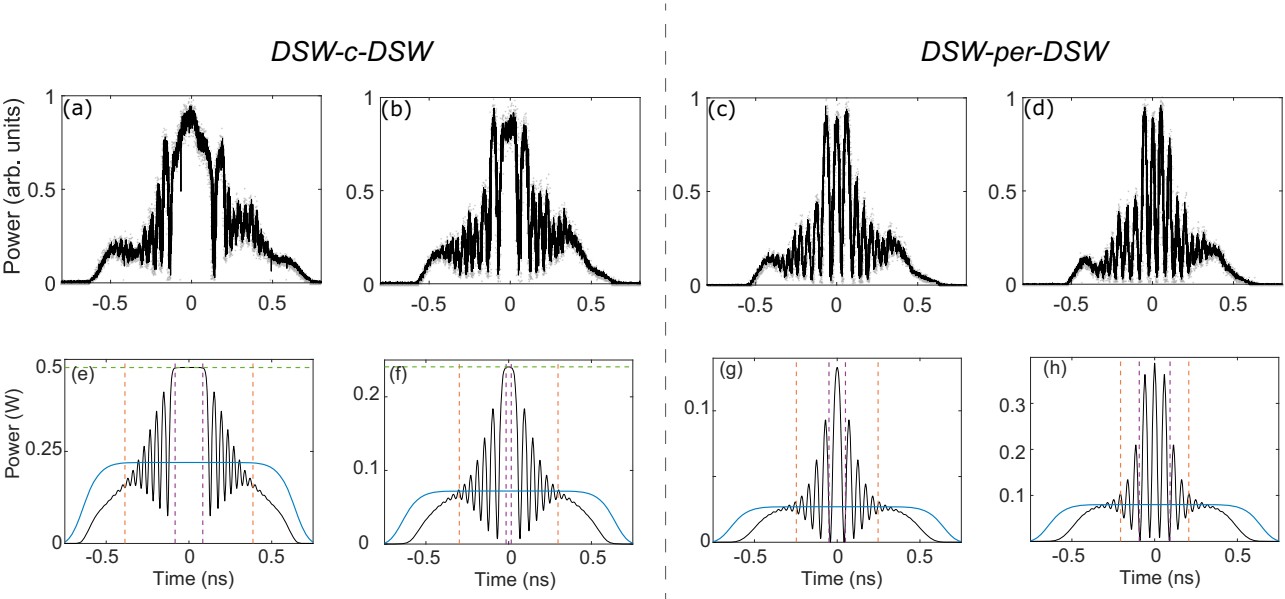

**Fig. 6 Pushing piston experiment.** Shock formation: DSW-c-DSW (**a, b**) and their phase transition to DSW-per-DSW (**c, d**), observed for $\Delta f_0 = 9.4$ GHz, and power: (**a**) $P_0 = 220$ mW ($u_0 = 0.95$); (**b**) $P_0 = 72$ mW ($u_0 = 1.66$); (**c**) $P_0 = 27$ mW ($u_0 = 2.71$); (**d**) $P_0 = 8$ mW ($u_0 = 4.97$). **e–h** Output profiles obtained from numerical simulation of the NLSE (1). Vertical dashed lines arise from modulation theory: linear DSW edges (orange) and soliton edges in (**a, b**) or periodic wave edges in (**c, d**) (magenta). In (**e, f**) the horizontal green line stands for $\rho_i P_0$, with $\rho_i$ from Eq. 4.

trace show a distortion, which is due to the non-perfectly flat plateau of the injected modulated pulse (see Fig. 4e). The hole is progressively dug during propagation until, in the case at power $P_0 = 165$ mW, the bottom of the RWs touches on the intermediate or rarefied state at constant power $\rho_i P_0$ predicted by the dispersionless NLSE (see dashed green horizontal line in Fig. 5c). Conversely, at lower power $P_0 = 35$ mW, which corresponds to more negative $u_0$, while the intermediate state becomes considerably lower (dashed green in Fig. 5d), both the experiment and the simulation exhibit only a slightly darker hole compared with previous case. This is due to the fact that, lowering the power results also in a slower dynamics and a shorter effective length ($z_L = L\gamma P_0$). As a result, the two RWs are expected to finally dig to the constant rarefied state at power $\rho_i P_0$ only at distances which far exceed the actual fiber length $L = 15$ km. Conversely, when $u_0$ increases above zero, a transition to the state DSW-c-DSW occurs (cyan area in Fig. 3c). In our experiment, such transition can be observed by reversing the step in frequency to become descending as in Fig. 4b. The relative output traces obtained for two different powers (or normalized velocities) are shown in Fig. 6a, b. We clearly observe the formation of two nearly symmetric DSWs connected by a constant intermediate state, which, in contrast with previous case, marks the highest power of the waveform. The experimental traces compare well with the corresponding simulations of the full NLSE (1) reported in Fig. 6e, f, and performed with ideal input (modulated super-Gaussian pulses, see Methods). We also point out that the location of the linear and soliton edges of the DSWs exhibit a satisfactory agreement with the predictions of Whitham theory (Eq. 6), vertical dashed lines in Fig. 6e, f.

Importantly, when the intermediate state shrinks to zero the state DSW-c-DSW is no longer sustainable. According to Whitham theory this occurs at threshold $u_0^{th} = 2$, above which the two DSWs are connected through a periodic nonlinear wave (see Supplementary Information Note 2 for more technical details). We have checked quantitatively that this phase transition, which possesses no analogy in the realm of classical non-dispersive fluids, can be observed by decreasing further the

power in order to increase $u_0$ above threshold. The output traces relative to $P_0 = 27$ mW and $P_0 = 8$ mW ($u_0 = 2.71$ and $u_0 = 4.97$) are reported in Fig 6c, d. They clearly show the DSWs to be connected through a periodic wave instead of a constant state, in good agreement with the corresponding simulations displayed in Fig. 6g, h. Note that, in this case, the DSWs no longer possess soliton edges, but rather connect smoothly to the periodic wave at temporal locations that can be calculated by Whitham modulation theory (magenta dashed lines in Fig. 6g, h; technical details in Supplementary Information Note 2).

Finally, we have also tested the peculiar transition DSW-c-DSW to DSW-per-DSW (cyan and green areas in Fig. 3c) in the more general case where the Riemann initial data involve a jump both in frequency and power. We refer to this regime as the asymmetric case since this type of initial condition breaks the symmetry between the left-going and right-going DSWs. In terms of analogy with gas dynamics this initial datum would correspond to a mixed case where two canonical initial conditions of the piston type (pure jump in velocity) and shock tube type (pure jump in density) are combined together. However, in common facilities for standard fluids, such as gas dynamics tube experiments or shallow water tanks, this is a very challenging task, and we are not aware of experimental results obtained for such case. The result of the experimental characterization of the input is shown in Fig. 7a, b. In this case, it is clear that the positively and negatively chirped portions have strongly different intensity. As shown, we illustrate the regime characterized by a large extinction ratio $\rho_0 = P_R/P_L = 0.15$, because this guarantee that cavitation takes place.

Figure 7c, d report the temporal power profiles observed at the output of the DCF, when operating at constant jump in frequency ($\pm 9.7$ GHz), constant extinction ratio $\rho_0 = 0.15$, and a variable power $P_0$ in order to vary the normalized chirp $u_0$. At $P_0 = 240$ mW, which corresponds to a normalized chirp $u_0 = 0.94 < u_0^{th} = 1.39$ the evolution is expected to give rise to DSW-c-DSW. This is shown in Fig. 7c, where we clearly observe the constant state (plateau) separating two strongly asymmetric DSWs. In particular the right DSW exhibits a cavitation point falling within its envelope. When the power is decreased to

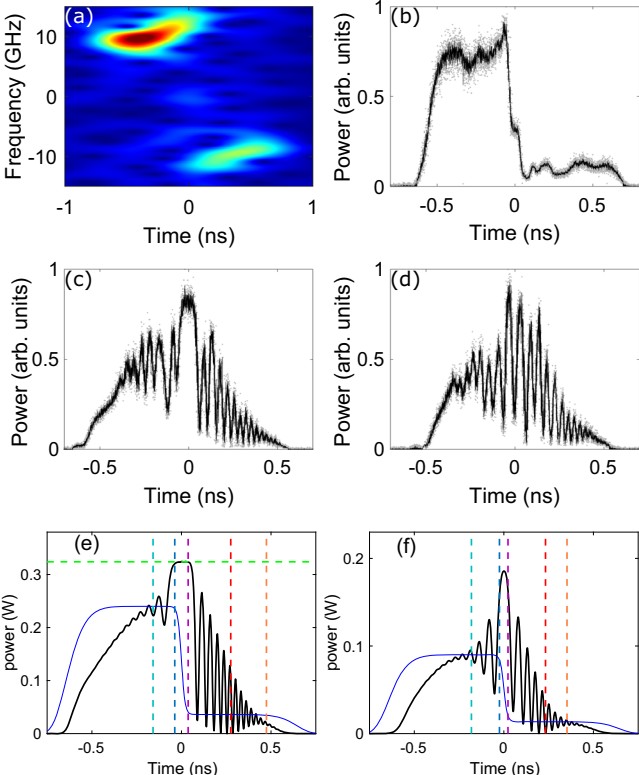

**Fig. 7 Mixed piston—shock tube problem.** Transition from DSW-c-DSW to DSW-per-DSW in the asymmetric case, with fixed $\Delta f = 9.7$ GHz and additional jump in power with nominal extinction ratio $P_R/P_L = 0.15$: (**a**) spectrogram of the input; (**b**) profile of input power jump; (**c, d**) output power profiles: (**c**) $P_0 = 240$ mW ($u_0 = 0.94$), below threshold; (**d**) $P_0 = 90$ mW ($u_0 = 1.52$), above threshold. **e, f** Output profiles obtained from numerical simulation of the NLSE (1). The vertical dashed lines stand for the vacuum point of the R-DSW (red), and the edge velocities of the DSWs (orange and cyan for linear edges, blue and purple for soliton or inner edges). In (**e**) the horizontal green dashed line stands for $\rho_i P_0$.

$P_0 = 90$ mW, which correspond to the above threshold value $u_0 = 1.52 > u_0^{th}$ the constant state disappears and the two DSWs appear to be connected by a periodic wave, as shown in Fig. 7d.

In Fig. 7e, f, we report the corresponding NLSE simulations with ideal initial data (blue curves in the figure). The agreement with the observed profiles is reasonably good. We attribute the discrepancies to the imperfect flatness of the input power states displayed in Fig. 7b, and to small deviations from the perfect temporal synchronization of the chirp and power steps, which is difficult to quantify. Importantly note that Whitham theory allows to predict with very good accuracy the location of the vacuum point in both regimes. Conversely, since the DSWs become strongly asymmetric, modulation theory allows to predict with good accuracy the temporal location of the edges (see orange and purple dashed vertical lines in Fig. 7e, f) of the DSW-R which is quite extended in time, whereas it is less accurate for the DSW-L, which is substantially narrower.

## Discussion

In summary, we have fully characterized the phase transitions associated with the Riemann problem in a fluid of light whose behavior is ruled by the universal NLSE. The physical piston is replaced by a stepwise optical pulse over which a quasi-instantaneous frequency chirp is imprinted, allowing to reproduce any velocity-density pair input conditions which mimic problems that span from the pure piston problem to its mix with the shock tube problem. These

specially designed pulses are launched in a defocusing optical fiber with actively compensated losses. In this way, the system is modeled by integrable NLSE and quantitative comparisons with theoretical developments from the Whitham modulation theory can directly be performed with experiments. We have been able to report: (i) a comprehensive study of the phase transitions that occur in the dispersive piston problem ruled by the defocusing NLSE; (ii) the observation of a regime, which has no similarity in gas dynamics, featuring two DSWs connected through an unmodulated periodic wave; (iii) the observation of asymmetric DSWs and their critical transition to the fully undulatory solution that follows from the most general Riemann problem involving a simultaneous jump both in power and chirp. All these observations are in very good agreement with theoretical predictions and numerics. This confirms that fiber-based optical systems are peerless testbeds to investigate the extension of gas dynamics problems to superfluid regimes by taking benefit of the analogy between optics and fluid-dynamics supported by the universality of the NLSE.

## Methods

**The dispersive Riemann problem.** A step input in power (density) and velocity (chirp) ruled by the NLSE evolves into several different combinations of RWs and DSWs connected by constant or periodic states. In real world units, the classification of the dynamics depends on four arbitrary parameters, namely the boundary values across the chirp and power jumps. However, using normalized variables $\rho$ and $u$, without loss of generality, we can assume, initial conditions characterized by only two parameters $(\rho_0, u_0)$:

$$\rho(t,0) = 1 + (\rho_0 - 1)\theta(t); \quad u(t,0) = u_0 - 2u_0\theta(t), \qquad (9)$$

where $\theta(t) = [1 + \text{sign}(t)]/2$ is the Heaviside unit step function. Equation 9 implies a left to right symmetric jump in velocity, from $u_L = u_0$ to $u_R = -u_0$, accompanied by a power jump from $\rho = \rho_L = 1$ to $\rho = \rho_R = \rho_0$. Non-symmetric input jump in $u$ only introduces a net drift in the problem, whereas the step in $\rho$ can always be normalized to have $\rho = 1$ over one of the boundaries, that we choose to be the left side. This allows us to classify all the possible evolution scenarios in a simple parameter plane $(\rho_0, u_0)$. According to the general theory of $2 \times 2$ conservation laws[23], the decay of the step-initial data ruled by the dispersionless NLSE can occur through the generation of a pair of fundamental waves, each being of the shock or rarefaction type, separated by a constant state. Therefore, three possible combinations can emerge: (i) CSW-CSW, (ii) CSW-RW, (iii) RW-RW, depending on the value of the initial data $(\rho_0, u_0)$.

In the dispersive regime ruled by the NLSE, however, the solution of the same problem becomes more challenging since it requires to resort to Whitham modulation theory, which describes a potentially more rich dynamics. Following pioneering results[27,55,56] and our calculations outlined in Supplementary Information Note 2, the result of such an approach is conveniently summarized in Fig. 3, where we report in the plane $(\rho_0, u_0)$ the domains where different wave pairs are expected to emerge. Five different regimes are highlighted by domains of homogeneous color in Fig. 3c, whereas curves that separate the domains denote dynamic transitions among the different regimes. We also display in Fig. 3a, b, d, e, typical output power profiles of the different decay scenarios, as obtained from Whitham modulation theory. The parameter values of such examples, as well as those of the experimental data are highlited in the parameter plane in Fig. 3c by green circles and blue crosses, respectively.

We also emphasize that the left $(\rho_0 < 1)$ and right $(\rho_0 > 1)$ white domains describe exactly the same physics, differing only for the direction of expansion of the DSW and RW pair. More precisely, the results in the semi-plane $\rho_0 > 1$ can be mapped in the semi-plane $\rho_0 < 1$ with the transposition $\rho_0 \rightarrow 1/\rho_0$ and $t \rightarrow -t$. For completeness, in Supplementary Fig. 5 reproduces Fig. 3 with additional results from ref. [32], which has allowed for a quantitatively accurate characterization of the cavitation which appears along the DSW when crossing above the dashed gray curve.

**Numerical simulations.** The numerical simulations in Figs. 5–7 as well as in Fig. 2a–c, are performed by integrating the NLSE with the pseudo-spectral split-step (Fourier) method. We modeled the stepwise input variation of the chirp as $\Delta\omega(T) = \pm T_0^{-1} u_0 \tanh(T/W_0)$, where the upper (lower) sign stands for ascending (descending) step, yielding the RW (DSW) pair. In all simulations we set $W_0 = 20$ ps, which is representative of the estimated rise time $T_r \sim 50$ ps (10 to 90 % of the final state) in the experiment. This chirp is imprinted on a flat-top pulse with super-Gaussian power profile $P_{pulse}(T) = P_0 \exp[-(T/W_G)^{10}]$ (blue curves in Figs. 5–7), with duration $W_G = 700$ ps. In the asymmetriic case shown in Fig. 7, we further embed in the flat-top pulse (around $T = 0$) a stepwise variation of power modeled as $P_{step}(T) = P_0\left[\rho_0 + (1 - \rho_0)\frac{1}{2}[1 - \tanh(T/W_0)]\right]$, where $W_0 = 20$ ps, and the extinction ratio is fixed to $\rho_0 = 0.15$. We remark few points concerning the simulations: (i) first, we did not make use of any adjustable (fitting)

parameter in the numerics; (ii) second, the additional Riemann-like problems associated with the sharp edges of the flat-top pulses do not interfere with the dynamics discussed in the paper. Indeed such initially abrupt tails only smooth out upon propagation (as observed in the numerics), while they cannot produce shocks due to their decay to zero, as discussed in detail in[32]; (iii) the overshoot and the tiny oscillations around the upper edges of the RWs in Fig. 5a, b are not a numerical artifact, but rather a manifestation of Gibbs phenomenon[32,62]. The experimental observation of these very small-scale oscillations, however, is hampered by the insufficient uniformity of the modulated flat-top pulses visible in Fig. 4e.

**Experimental setup**. The setup has been specifically designed to face two key challenges: (i) to impress a consistent and rapid frequency modulation on the input signal; and (ii) to compensate for the fiber loss in the main fiber (DCF). The frequency modulation should be strong enough to produce frequency shifts which are as large as $\Delta f_0 \sim \pm 10$ GHz (though, relative to the carrier $f_0 = 192.18$ THz, the variation $|\Delta f_0|/f_0 \sim 5 \times 10^{-5}$ remains small, as usual in optics), over relatively long duration (about 0.5 ns, which is long enough to observe the full development of DSW envelopes typically made of tens of modulation periods). Moreover, the transition to switch from negative to positive values of frequency and vice versa, must be ultra-fast (typical rise time ~ 50 ps) so to approximate the ideally instantaneous stepwise variation. To this end we resort to an all-optical method based on cross-phase modulation (XPM) as in ref. [63] in order to be able to generate these large frequency chirps. This corresponds to a maximum phase value of $10\pi$, well above the typical $\pi$ rad., the characteristic maximum value accessible with standard phase modulators used in telecom applications. The main idea behind the preparation of the Riemann-like input (point (i)) is to use two laser sources at slightly different wavelengths suitably modulated in amplitude, which are combined and injected in a first fiber, namely a highly nonlinear fiber (HNLF). The HNLF has the key role of transforming the amplitude modulation of one of the laser sources into the frequency modulation of the other source via XPM. The two lasers are independently modulated as sketched by the two main blocks denoted in Fig. 4 as "Phase modulation branch" and "Square branch", respectively. In particular, in the lower block (Square branch) we make use of a continuous laser diode emitting at $\lambda_2 = 1561$ nm (which fixes the carrier pulsation introduced in the text to $\omega_0 = 2\pi c/\lambda_2$), which is intensity modulated by an electro-optic modulator (EOM) driven by an arbitrary waveform generator (AWG2), amplified in an Erbium-doped fiber amplifier (EDFA) and spectrally filtered to remove the amplified spontaneous emission in excess. The intensity modulation produces a train of square pulses with 25 MHz repetition rate. The pulses provide constant power (density) over 2 ns duration, which is expected to be a sufficiently wide temporal window for the DSWs to develop in the piston-like experiment. Conversely, the upper block (Phase modulation branch) is devoted to impress a proper phase modulation to the beam at $\lambda_2$. To this end we start from a continuous laser diode emitting at $\lambda_1 = 1539$ nm, and impress an intensity modulation via a second EOM driven by AWG1, after which the signal is again amplified and filtered. The modulation is synchronous with that of the other block, whereas the intensity waveform can be chosen to be either M-shaped (sketched in red in Fig. 4a) or triangularly shaped (sketched in green in Fig. 4a). The output of the two blocks are combined through a 90:10 coupler, so that we launch in the HNLF a dual wavelength signal constituted by square pulses with flat phase with typical peak power $P_2 = 40$ mW superimposed to a more powerful beam (peak $P_1(T)_{max} = 7$ W) of different color and suitable temporal shape (either M-like or triangular). The linear polarization of the two beams is controlled to be parallel in order to maximize the effect of XPM. During propagation in the HNLF, the beam at $\lambda_1$ induces via XPM an output phase modulation $\phi_2(t) = 2\gamma_H L_H P_1(T)$ over the beam at $\lambda_2$, $\gamma_H = 12$ (W km)$^{-1}$ and $L_H = 500$ m being the nonlinear Kerr coefficient and the length of the HNLF, respectively. This corresponds to a chirp $\Delta\omega(T) = -d\phi_2(T)/dt = -2\gamma_H L_H dP_1(T)/dT$, and in turn to a profile of the equivalent initial gas velocity $u(t) = \Delta\omega(T)T_0 = 2\pi\Delta f(T)\sqrt{k''/(\gamma P_0)}$. The abrupt change of slope in the M-shaped or triangular intensity modulation is converted, due to the derivative that links the frequency to the phase, into a step-like variation of the chirp or equivalent fluid velocity, which is either descending (for the M-shaped case) or ascending (for the triangular shape), as sketched in Fig. 4a. At the output of the HNLF the beam at $\lambda_1$, as well as the multiple sidebands produced by four-wave mixing of the input beating between $\lambda_1$ and $\lambda_2$, are filtered out through a bandpass filter. The remaining, strongly chirped beam at $\lambda_2$ constitutes the Riemann-like input which is injected in the main fiber, i.e., the $L = 15$ km long DCF. In such fiber it becomes crucial to compensate the losses (point (ii)) which amounts to 0.5 dB/km or nearly 80% total loss. This is performed by exploiting the Raman gain from a counterpropagating pump at $\lambda_2 = 1480$ nm (for more details see ref. [32]). Finally, the output of the DCF is monitored both spectrally, by means of an optical spectrum analyzer (OSA) and in time domain by means of an optical sampling oscilloscope (OSO) synchronized by the clock of the two AWGs.

## Data availability
Most of the relevant data used in this paper are contained in the Supplementary Information, while further data are available from the corresponding authors upon reasonable request.

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

## Acknowledgements

The present research was supported by IRCICA (USR 3380 CNRS, project IRCICA 2020), Agence Nationale de la Recherche (Program Investissements d'Avenir, I-SITE VERIFICO); Ministry of Higher Education and Research; Hauts de France Council; European Regional Development Fund (Photonics for Society P4S, FUHNKC, EXAT, FELANI). The authors are grateful to L. Bigot, E. Andresen, and IRCICA-TEKTRONIX European Optical and Wireless Innovation Laboratory for technical support about the electronic devices. Discussions with A. Kamchatnov, N. Pavlov, and P. Sriftgizer are gratefully acknowledged. The authors thanks T. Sylvestre for providing the HNLF fiber. S.T. acknowledges support from the Italian Ministry of University (MUR), grant PRIN-2020X4T57A.

## Author contributions

A.B., G.X., A.K., and A.M. performed the experiments. S.T. and M.C. developed the theory and realized the numerical simulations. All authors contributed to analysing the data and writing the paper. A.B. contributed equally to this work with G.X.

## Competing interests

The authors declare no competing interests.
