## [Peer Review File · Nature Communications]

The piston Riemann problem in a photon superfluidREVIEWER COMMENTS

Reviewer #1 (Remarks to the Author):

The work of Bendahmane et al. explores experimentally the generalized piston problem of fluid mechanics by implementing it on the hydrodynamic-like equations that arise when a properly designed light pulse propagates inside a nonlinear fiber after imposing a step-like variation of velocity and density as initial conditions.

The topic is certainly timely and could potentially be of interest for a broad audience.

However, we feel that the manuscript in its present form is very hard to understand as it uses a lot of very field-specific technical vocabulary.

A clear idea should be put forward to highlight the key results of this research (even at the cost of being more concise or sacrificing some of the technicalities to allow for a better general understanding of the challenges).

As is, the manuscript is better suited for a more specialised journal with a target audience that is going to be familiar with the specificities of the field. The current version of the manuscript is too technical for the general audience of Nature Communication.

We strongly encourage the authors to revise the manuscript, rewriting it in a more pedagogical form.

Some examples (not exhaustive) of what should be amended

-Eq. (1). There is no explanation whatsoever of what the parameters are : Z , k , γ , T

-Fig. 1 : it conveys too much information, consider the option to simplify or split in separated figures

-Sentences like the following: "This indeed reproduces the well-known case study known as shock tube problem in gases".

- The authors often talk about "a phase transition to new regime occurs" p4 and p7 "we have fully characterized the phase transitions associated with the Riemman problem". Could they precise which kind of phase transitions they are dealing with? Between which regimes?

Reviewer #2 (Remarks to the Author):

The paper presents an experimental study of chirped, high power, broad pulses that are input to a fiber with normal dispersion. Using some rather sophisticated techniques, the chirp is prepared to rapidly transition between two nearly uniform-in-time values at the center of the pulse. The input power across the middle of the pulse exhibits some variation (~10-15%). The output power is reported at the end of a 15 km fiber. The fiber output is theoretically interpreted using the paraxial approximation and the lossless defocusing nonlinear Schrodinger equation (NLSE). Numerical simulations of the NLSE exhibit qualitative agreement with the observed output power profiles. By invoking the transformation of the NLSE's complex light envelope to power-chirp variables (the Madelung transformation), the results are interpreted in a dispersive hydrodynamic context in terms of dispersive shock waves and rarefaction waves. This is possible due to the hydrodynamic-like conservation laws of the NLSE. With this interpretation in hand, the authors present their experimental results through the lens of the hydrodynamic piston problem, which is significantly altered from the classical gas dynamics case by the presence of the fiber's normal dispersion. Oscillatory features associated with dispersive shock waves, nonlinear periodic waves, and rarefaction waves are accessed by changing the input peak power while holding the frequency chirp jump the same.

These optical experiments are impressive and represent an important contribution to the field of dispersive hydrodynamics in nonlinear physics. While suitable theory for this dispersive piston problem and the more general Riemann problem (sharp input transition in chirp and power) has been around for quite some time, related experimental studies in optics, superfluids and fluid dynamics have focused on the case of a sharp transition in power alone (density/water height in the case of superfluids/shallow water waves). The experimental innovation of being able to precisely input a jump in the chirp (fluid velocity in superfluids/shallow water) allows the authors to probe new, intriguing dynamical regimes by changing the input peak power. In particular, they demonstrate convincing evidence for the generation of dispersive shock waves connected by a region of nearly constant power or a region consisting of a nonlinear, locally periodic wave. Leveraging the approximate symmetry of the optical input (even in power, odd in chirp), this result, through a "method of images" type argument, can be interpreted as a piston problem that generates a dispersive shock wave with a constant or periodic "wake". This bifurcation in dynamics is drastically different from the classical piston problem in gas dynamics for which there is only a shock and a constant wake. It highlights the novel features of hydrodynamic and hydrodynamic-like dynamics in media with very weak dissipation relative to wave dispersion. This is a laudable result and, in my opinion, deserves publication in a high profile journal such as Nature Comm. The following are some comments and questions for the authors to consider.

1. Why don't the authors report the output power in watts rather than arbitrary units? The numerical simulations are reported in dimensional units. Can't the power scale be determined or at least fitted across all the trials?

2. How fast does the initial chirp change? This could significantly affect the rarefaction wave profiles. The authors refer to rarefaction waves as smooth but, in fact, there are dispersive oscillations that can be quite large if the initial transition is sharp enough. The retracting piston rarefaction wave experiments reported in Fig. 4a,b exhibit oscillatory features. The authors argue that these distortions are due to the nonuniformities in the initial pulse power. Why couldn't these also be due to dispersion?

3. What is the input chirp profile for the numerical simulations? Has any sort of fitting or shaping been performed in order to obtain the observed agreement between experiment and simulation? It is essential to report the numerical input and how it relates to the experimental input.

4. Since the input chirp profile was measured, can the output chirp profile also be measured? If not, it would be helpful to report the input/output chirp from numerical simulations.

5. The outermost edges of the pulse also exhibit a sharp transition to zero power. Can the authors comment on this Riemann problem?

6. Why not use the actual measured input power and chirp as input to the numerical simulations?

7. Why does the observed dispersive shock wave harmonic edge move consistently faster than the numerical simulation? Do the authors expect higher order effects to play a role here?

8. On page 7, the authors describe the asymmetric Riemann problem as "two pistons pushing on two sectors". What are sectors? I am unable to understand this analogy so a refined description would be helpful.

9. In the abstract and intro, the authors refer to 2-shock, 2-DSW, 2-rarefaction, etc but never define these terms. In the body of the text and in Fig. 1, the authors use the terms DSW-L and DSW-R, which presumably would be 1-DSW and 2-DSW, respectively. It is recommended to use one terminology, not both.

10. On page 3, the authors refer to the DSW soliton edges as "leading". For this positive dispersion case, the soliton edge is trailing because the soliton edge in DSW-R is the slowest and trails the faster harmonic edge.

11. Please report the carrier frequency so that the chirp excursion reported on page 5 can be interpreted in a relative sense.

12. In the supplement, the authors use the dispersionless model to obtain shock curves in eq. (S4) and wave curves eq. (4) in the main text relating left and right states between classical shocks and rarefaction waves. While it is true that the shock and wave curves approximately agree to third order in the jump height (this is a general result from conservation law theory, see, e.g., Courant and Friedrichs 1948), it is important to note that they are not simultaneously admissible. In particular, the shock curve connecting two states requires an entropy condition to be satisfied for an admissible shock whereas the wave curve requires a certain monotonicity condition (increasing or decreasing, depending on the 1 or 2 wave). So, while the wave and shock curves approximately agree, only one of them is admissible. The important feature of NLSE dispersive hydrodynamics is that the wave curve applies in both cases. When admissible as a simple wave solution of the dispersionless equations, a rarefaction wave is generated but when not admissible, a dispersive shock wave is generated. In both cases, the wave curve holds.

13. In Fig. S4, the caption and figure labels (Ref [6] and Ref [8]) do not agree.

Reviewer #3 (Remarks to the Author):

In the manuscript titled: "*The piston Riemann problem in a photon superfluid*," the authors use a photonic platform based on nonlinear optical fiber to detect extreme hydrodynamic states of the light through exploring the photonic analog of the generalized piston problem in gas dynamics. Phase transitions associated with the Riemann problem in a fluid of light are fully characterized theoretically and experimentally by considering the dispersive regime ruled by defocusing NLSE, where an appropriate shaping of the power and frequency chirp profiles of the optical pulse at the onset of a DCF allows reproducing any density-velocity pair input condition. The authors reported a comprehensive investigation in the temporal domain of the phase transitions that occur in dispersive piston problems, in analogy with the gas-dynamics counterpart. Remarkably, for high values of the frequency chirp, the observation of new regimes that have no similarity in gas dynamics such as two dispersive shock waves connected through an unmodulated periodic wave has been also reported. **In my opinion, the scientific content of this manuscript brings about a high degree of novelty, useful for the scientific community in optics and photonics. Theoretical concepts are developed in a detailed manner and enough comprehensible, even for an audience not only the expertise of this field. The claim is well supported by numerical and experimental results. Figures and caption descriptions are clear and complete. From my point of view, the impact in terms of the**

novelty of this research work can satisfy the requirements for publication in the Nature Communications journal.

I would suggest the authors address some minor revisions and comments

1. Some experimental results in the manuscript present some slight distinctions when compared to corresponding numerical simulations. Could they explain why predominant lobe powers were no longer observed as the temporal stepwise chirp increases for the output traces plotted in Figs. 5 and 6?
2. In the experimental setting, the authors mentioned that the loss in the DCF fiber was compensated by a counter-propagating Raman pump. Did they also compensate for high-order dispersions (i.e. TOD) and high nonlinear effects (i.e. stimulated Raman Scattering, self-steepening)? Which is the most limiting factor for the experiment?
3. Could the authors give more details on how the coefficient of pondering of the cost function F_{factor} was chosen in their compensating method to estimate the degree of fitness J ? Did they compensate even the triangular driving signal used in the experimental setup.
4. Is it possible by employing the pulse shaping technic based on cross-phase modulation to impress an asymmetric step-like chirp?
5. In the manuscript is mentioned that the experimental square pulse provides a constant power over 2ns duration and 25MHz repetition rate, and in the normalization of the NLSE, the parameter T_0 scale with power and is not directly linked to the initial pulse duration in order to control the magnitude on the chirp. For the sake of clarity, it would be better to report the actual pulse duration of the onset square pulse used in simulations modeling the experimental results.
6. In the caption of Fig. S4, the authors reported red stars as the experimental results from Ref. [6], while in panel S4(d) the same results are referred to Ref. [8]. Furthermore, in the same panel, the blue cross results indicate different figures with respect to the Fig. 2 of the main text. I suggest checking it.
7. Panels (a) and (b) of Fig. 2. Which is the input condition of the power density for these two examples of RW-c-DSW regime? Is it an asymmetric case with a stepwise profile?
8. In the animation provided by the authors, y- and x-axis labels in the frequency chirp plot should be changed with u_0 and ρ_0 to be consistent with Fig. 2 in the main text of the manuscript and Fig. S4 in the supplementary.

9. I suggest authors revise the reference list of their manuscript because the abbreviation form of some cited journals is not reported in the correct manner. For the sake of completeness, I would also suggest them cite along with Ref. [34] even the following paper dealing with temporal Riemann pulse phenomena in nonlinear optical fibers:

S. Wabnitz, "Optical tsunamis: shoaling of shallow water rogue waves in nonlinear fibers with normal dispersion," *J. Opt.* **15**, 064002 (2013)

D. Bongiovanni, *et al.*, "Third-order Riemann pulses in optical fibers," *Opt. Express* **28**(26), 39827-39840 (2020)

Reviewer #1 (Remarks to the Author):

The work of Bendahmane et al. explores experimentally the generalized piston problem of fluid mechanics by implementing it on the hydrodynamic-like equations that arise when a properly designed light pulse propagates inside a nonlinear fiber after imposing a step-like variation of velocity and density as initial conditions. The topic is certainly timely and could potentially be of interest for a broad audience.

However, we feel that the manuscript in its present form is very hard to understand as it uses a lot of very field specific technical vocabulary.

Authors: We thank the Referee for recognizing that the paper is certainly timely and of potential interest for a broad audience. We have further revised the paper to limit as much as possible the technical language which would be specific to a particular field.

A clear idea should be put forward to highlight the key results of this research (even at the cost of being more concise or sacrificing some of the technicalities to allow for a better general understanding of the challenges).

Authors: We have expanded the last paragraph of the intro to better highlight the challenges that our experiment faces. We believe that we have clearly stated the goals and key results: to realize for the first time in any quantum fluid the (dispersive) piston problem and its Riemann problem generalization and to detect a new wave regime (shocks connected through a nonlinear periodic wave) that occurs above a velocity threshold.

As is, the manuscript is better suited for a more specialised journal with a target audience that is going to be familiar with the specificities of the field. The current version of the manuscript is too technical for the general audience of Nature Communication.

Authors: We respectfully disagree on the fact that the paper would be too technical for the general audience of Nature Commun. In the paper, we presented the main ideas and concepts from the phenomenological point of view in self-consistent way, shifting to "Methods" the more technical methodological content, and to Supplemental Information the more specific technical details (which does not necessarily need to be read to understand the main text). Comparing our manuscript with other papers published in Nature Commun. in the same field (specifically Refs. 12,20,30,36 of the revised text), our strong feeling is that our paper is by no means more technical or more oriented to a smaller community of experts. Finally, we remind that, contrary to Reviewer 1, Reviewer 3 clearly underlined that our work is accessible also to non-experts in the field by writing: *"...concepts are developed in a detailed manner and enough comprehensible, even for an audience not only the expertise of this field"*

We strongly encourage the authors to revise the manuscript, rewriting it in a more pedagogical form.

Authors: We have followed the advice of the Referee, in spite of what premised in the previous point. We have further revised the paper in order to avoid any jargon which is specific to our field (nonlinear optics) or the field of gas dynamics which is the term of comparison here.

Some examples (not exhaustive) of what should be amended

Authors: Below we address the specific examples made by the Referee

-Eq. (1). There is no explanation whatsoever of what the parameters are: Z , k , γ , T

Authors: We thank the Referee for pointing this out. We have included such explanation in the main text, after Eq. (1), while more details are available in "Methods".

-Fig. 1 : it conveys too much information, consider the option to simplify or split in separated figures

Authors: we have split the figure into two distinct figures, as suggested, the first one being a simple and easily understandable cartoon-like picture that express how our experiment is conceived.

-Sentences like the following: "This indeed reproduces the well-known case study known as shock tube problem in gases".

Authors: we have expanded by explaining what the shock wave tube problem means (Pag.4, bottom of the first column).

- The authors often talk about "a phase transition to new regime occurs" p4 and p7 "we have fully characterized the phase transitions associated with the Riemann problem". Could they precise which kind of phase transitions they are dealing with? Between which regimes?

Authors: A phase transition occurs anytime the dynamics qualitatively change as a consequence of a variation of the initial jump parameters (ρ_0 , u_0) such that one of the boundaries in Fig. 3(d) is crossed [formerly Fig. 2(d)].

From now on we refer to the numbering of the revised version]. We have better specified this in the description of Fig. 3, and throughout the paper.

Reviewer #2 (Remarks to the Author):

The paper presents an experimental study of chirped, high power, broad pulses that are input to a fiber with normal dispersion. Using some rather sophisticated techniques, the chirp is prepared to rapidly transition between two nearly uniform-in-time values at the center of the pulse. The input power across the middle of the pulse exhibits some variation (~10-15%). The output power is reported at the end of a 15 km fiber. The fiber output is theoretically interpreted using the paraxial approximation and the lossless defocusing nonlinear Schrodinger equation (NLSE). Numerical simulations of the NLSE exhibit qualitative agreement with the observed output power profiles. By invoking the transformation of the NLSE's complex light envelope to power-chirp variables (the Madelung transformation), the results are interpreted in a dispersive hydrodynamic context in terms of dispersive shock waves and rarefaction waves. This is possible due to the hydrodynamic-like conservation laws of the NLSE. With this interpretation in hand, the authors present their experimental results through the lens of the hydrodynamic piston problem, which is significantly altered from the classical gas dynamics case by the presence of the fiber's normal dispersion. Oscillatory features associated with dispersive shock waves, nonlinear periodic waves, and rarefaction waves are accessed by changing the input peak power while holding the frequency chirp jump the same.

These optical experiments are impressive and represent an important contribution to the field of dispersive hydrodynamics in nonlinear physics. While suitable theory for this dispersive piston problem and the more general Riemann problem (sharp input transition in chirp and power) has been around for quite some time, related experimental studies in optics, superfluids and fluid dynamics have focused on the case of a sharp transition in power alone (density/water height in the case of superfluids/shallow water waves). The experimental innovation of being able to precisely input a jump in the chirp (fluid velocity in superfluids/shallow water) allows the authors to probe new, intriguing dynamical regimes by changing the input peak power. In particular, they demonstrate convincing evidence for the generation of dispersive shock waves connected by a region of nearly constant power or a region consisting of a nonlinear, locally periodic wave. Leveraging the approximate symmetry of the optical input (even in power, odd in chirp), this result, through a "method of images" type argument, can be interpreted as a piston problem that generates a dispersive shock wave with a constant or periodic "wake". This bifurcation in dynamics is drastically different from the classical piston problem in gas dynamics for which there is only a shock and a constant wake. It highlights the novel features of hydrodynamic and hydrodynamic-like dynamics in media with very weak dissipation relative to wave dispersion. This is a laudable result and, in my opinion, deserves publication in a high profile journal such as Nature Comm. The following are some comments and questions for the authors to consider.

Authors: We thank the Referee for considering our experiment impressive and an important contribution to the field of dispersive hydrodynamics, and finally for suggesting publication in a high profile journal like Nature Communications. Below we reply to the specific comments and questions (corresponding changes in the paper are highlighted in red).

1. Why don't the authors report the output power in watts rather than arbitrary units? The numerical simulations are reported in dimensional units. Can't the power scale be determined or at least fitted across all the trials?

Authors: We did not measure the absolute power as a function of time at the fiber output. While we could normalize the traces with respect to the simulated output power, we find more fair to leave the experimental figures in arbitrary units (relative power). In any case, the aim of the figures is to furnish a visual, qualitative, comparison of the experimental and numerical traces, which is given by both options.

2. How fast does the initial chirp change? This could significantly affect the rarefaction wave profiles. The authors refer to rarefaction waves as smooth but, in fact, there are dispersive oscillations that can be quite large if the initial transition is sharp enough. The retracting piston rarefaction wave experiments reported in Fig. 4a,b exhibit oscillatory features. The authors argue that these distortions are due to the nonuniformities in the initial pulse power. Why couldn't these also be due to dispersion?

Authors: We agree that dispersive oscillations can also be present for the rarefaction waves, due to dispersion, as known e.g. from asymptotic theory of dispersive PDEs (e.g. Leach & Needham, Nonlinearity 21, 2391 (2008) for KdV equation). This phenomenon has been discussed also in the framework of the Gibbs phenomenon for

PDEs with discontinuous initial data, which we now quote in Ref. [63]. The typical oscillations and overshoot expected in the experiment are indeed visible mostly around the high density edges of the rarefaction in the simulations in Fig. 5(c,d) of the revised version (performed with input chirp shape $\tanh(T/W_0)$, $W_0=20$ psec, realistic for the experiment, as we now report in detail in “Methods”, see next point). These oscillations are not a numerical artifact. However, as shown, they are tiny, and in the experiment the non-uniformity of the power profile (clearly visible in Fig. 4(e) of the revised version), hampers the clear observation of these small-scale oscillations.

3. What is the input chirp profile for the numerical simulations? Has any sort of fitting or shaping been performed in order to obtain the observed agreement between experiment and simulation? It is essential to report the numerical input and how it relates to the experimental input.

Authors: Thanks for the comment, we fully agree. In previous version, the numerical input chirp profile was reported in the caption of Fig. 4. In order to be more easily accessed, we have added in “Methods” a new subsection (“Numerical Simulations”), where we report all the parameters of the simulations. As stated in this subsection, we did not make use of any sort of fitting or shaping to better fit the experiment. Indeed, the simulations are made for pulses with ideal superGaussian power profile and tanh-shaped jump in frequency, and we have no adjustable (fitting) parameters in the numerical runs.

4. Since the input chirp profile was measured, can the output chirp profile also be measured? If not, it would be helpful to report the input/output chirp from numerical simulations.

Authors: The input chirp is measured through the spectrograms reported in Figs. 4(b,c) and 7(a) of the revised version, and explained in details in the Supplementary Information. At the output the chirp is expected to exhibit very fast variations over the same temporal scale of the power oscillations, as shown explicitly in Fig. 2(a) of the revised version. Our present method cannot resolve such fast variations. Indeed, resolving the psec-scale oscillatory structure of the chirp in an optical DSW would be a breakthrough by itself, which needs further work in the future.

As suggested, we have reported, in the Supplementary note 3, some other examples of the input/output numerical chirp, for the case of the most general Riemann input (jump in both power and chirp).

5. The outermost edges of the pulse also exhibit a sharp transition to zero power. Can the authors comment on this Riemann problem?

Authors: We thank the Referee for the comment. It is true that the sharp transition at the outermost edges of the pulse can be seen as an additional Riemann problem. However, the fact that the transition is towards a state with zero density (power) forbids the formation of shock. This is well known in hydrodynamics where this problem has one-to-one correspondence with the dam break problem in the so-called dry-bed case (zero density downstream), as already discussed in detail by some of us in Ref. [31]. The dynamics is trivial in this case, giving rise to rarefaction waves which describe the smoothing of the outermost pulse tails. We have inserted a brief comment on this point in the subsection “Numerical Simulations” in “Methods”.

6. Why not use the actual measured input power and chirp as input to the numerical simulations?

Authors: We understand the suggestion, and indeed we have performed also simulations by using the input extrapolated from measurements. We think, however, that it is more instructive to compare with numerics obtained under conditions which are closer to ideal ones.

7. Why does the observed dispersive shock wave harmonic edge move consistently faster than the numerical simulation? Do the authors expect higher order effects to play a role here?

Authors: We attribute the fact that, in the experiment, the harmonic edges of the DSWs look somehow enhanced to the fact that the power profile deviates from the ideally flat profile, and present external bumps where the harmonic edges develop, as clearly visible in Fig. 4(d) of the revised manuscript. We do not have evidence that higher-order effects (steepening term or intrapulse Raman) are important here.

8. On page 7, the authors describe the asymmetric Riemann problem as "two pistons pushing on two sectors". What are sectors? I am unable to understand this analogy so a refined description would be helpful.

Authors: We understand the doubt. By the word “sectors” we meant the two sections at different densities in a shock tube problem. We have rephrased the description.

9. In the abstract and intro, the authors refer to 2-shock, 2-DSW, 2-rarefaction, etc but never define these terms. In the body of the text and in Fig. 1, the authors use the terms DSW-L and DSW-R, which presumably would be 1-DSW and 2-DSW, respectively. It is recommended to use one terminology, not both.

Authors: Thanks for the recommendation, we now use a uniform terminology throughout the paper. Since we find rather arbitrary to identify the DSWs with numbers (1- and 2-), we maintain the notation DSW-L and DSW-R, while in the abstract/introduction, we simply refer to the whole wave pattern as a (rarefaction or shock) wave pairs.

10. On page 3, the authors refer to the DSW soliton edges as "leading". For this positive dispersion case, the soliton edge is trailing because the soliton edge in DSW-R is the slowest and trails the faster harmonic edge.

Authors: We stuck to fiber optics terminology, where the adjective "leading" is used for the front/tail that lies at earlier times being faster (in the sense that it travels a given space in lesser time). In this sense the soliton edge of the DSW-R is leading, and so it is the soliton edge of the DSW-L, when referred to the reversed direction of motion/expansion. However, we agree on the fact that also the opposite convention suggested by the Referee could be used, according to the fact that we interpret (in the spirit of shock wave theory) the quantities $\tau_{1,2}$ as velocities (dimensionally, they would represent inverse velocities). The two different conventions ultimately arise from the interchanged role of space and time in fiber optics versus shock wave theory or gas dynamics. Due to this ambiguity, we decided to remove the terms "leading" and "trailing", as they are not essential to understand.

11. Please report the carrier frequency so that the chirp excursion reported on page 5 can be interpreted in a relative sense.

Authors: As usual in optics, the carrier frequency is very large, so the relative variation is not very significant (at variance with microwaves, for instance). However, we agree that this can be a useful information in order to compare with other areas of interest (hydrodynamics, spintronics, BEC), so we have reported the value of the carrier and the relative variation in "Methods".

12. In the supplement, the authors use the dispersionless model to obtain shock curves in eq. (S4) and wave curves eq. (4) in the main text relating left and right states between classical shocks and rarefaction waves.

While it is true that the shock and wave curves approximately agree to third order in the jump height (this is a general result from conservation law theory, see, e.g., Courant and Friedrichs 1948), it is important to note that they are not simultaneously admissible. In particular, the shock curve connecting two states requires an entropy condition to be satisfied for an admissible shock whereas the wave curve requires a certain monotonicity condition (increasing or decreasing, depending on the 1 or 2 wave). So, while the wave and shock curves approximately agree, only one of them is admissible. The important feature of NLSE dispersive hydrodynamics is that the wave curve applies in both cases. When admissible as a simple wave solution of the dispersionless equations, a rarefaction wave is generated but when not admissible, a dispersive shock wave is generated. In both cases, the wave curve holds.

Authors: We thanks the Referee for the comment, we fully agree on that. The main purpose of Supplementary note 1 was to recall, on a more rigorous ground (compared with Fig. 1) the substantial identity between the compressive piston problem and the velocity Riemann problem in the dispersionless case. We also believe that it is important to mention the two possible approaches to classical shocks in this case, even if we agree on the fact that it is the simple-wave approach that matters for the dispersive case for both rarefactions and shocks. We have expanded on this point by adding an entire final paragraph to the end of Supplementary note 1, which reflects the comment by the Referee.

13. In Fig. S4, the caption and figure labels (Ref [6] and Ref [8]) do not agree.

Authors: We thank the Referee for noticing. After renumbering the References as initially meant, the correct Ref. is [8], and we have changed accordingly.

Referee 3

In the manuscript titled: "*The piston Riemann problem in a photon superfluid*," the authors use a photonic platform based on nonlinear optical fiber to detect extreme hydrodynamic states of the light through exploring the photonic analog of the generalized piston problem in gas dynamics. Phase transitions associated with the Riemann problem in a fluid of light are fully characterized theoretically and experimentally by considering the dispersive regime ruled by defocusing NLSE, where an appropriate shaping of the power and frequency chirp

profiles of the optical pulse at the onset of a DCF allows reproducing any density-velocity pair input condition. The authors reported a comprehensive investigation in the temporal domain of the phase transitions that occur in dispersive piston problems, in analogy with the gas-dynamics counterpart. Remarkably, for high values of the frequency chirp, the observation of new regimes that have no similarity in gas dynamics such as two dispersive shock waves connected through an unmodulated periodic wave has been also reported. **In my opinion, the scientific content of this manuscript brings about a high degree of novelty, useful for the scientific community in optics and photonics. Theoretical concepts are developed in a detailed manner and enough comprehensible, even for an audience not only the expertise of this field. The claim is well supported by numerical and experimental results. Figures and caption descriptions are clear and complete. From my point of view, the impact in terms of the novelty of this research work can satisfy the requirements for publication in the Nature Communications journal.**

Authors: We thank the Referee for positive evaluation of our work and for recommending it for publication in Nature Communications. Below we address the suggestions for minor revisions.

I would suggest the authors address some minor revisions and comments

1. Some experimental results in the manuscript present some slight distinctions when compared to corresponding numerical simulations. Could they explain why predominant lobe powers were no longer observed as the temporal stepwise chirp increases for the output traces plotted in Figs.5 and 6?

Authors: The predominant central lobe at low values of chirp u_0 in Figs. 5-6 (now Figs 6-7) is the signature that the two DSWs develop over a constant plateau (the top of the lobe). Increasing u_0 , the plateau shrinks until it disappears (at threshold given by Eq. (8)) and leave the two DSWs to be interconnected through a periodic unmodulated wave. We have better emphasized this key feature in the text.

2. In the experimental setting, the authors mentioned that the loss in the DCF fiber was compensated by a counter-propagating Raman pump. Did they also compensate for high-order dispersions (i.e TOD) and high nonlinear effects (i.e. stimulated Raman Scattering, self-steepening)? Which is the most limiting factor for the experiment?

Authors: We did not compensate for any higher-order effect. Third- and higher-order dispersion has negligible impact here since we do not operate close to zero dispersion wavelength. Although we cannot exclude that higher-order nonlinear effects (Raman, self-steepening) slightly impact the output power profiles, they have practically no effects on the thresholds in u_0 which characterize the changes of dynamics. In the experiment, the major discrepancies with the ideal case come from the distortion in the power profile [shown in Fig. 4(d,e) and Fig. (7)b of the revised manuscript], introduced by higher-order effects (spurious FWM processes or Raman effect involving laser 1 and 2) in the high-nonlinear fiber used to translate the amplitude modulation into the appropriate frequency chirp.

3. Could the authors give more details on how the coefficient of pondering of the cost function Ffactor was chosen in their compensating method to estimate the degree of fitness J? Did they compensate even the triangular driving signal used in the experimental setup.

Authors: The value for the pondering of the cost function F factor was chosen by a trial-and-error procedure in order to obtain a good convergence in both the shape of the pulse and its derivative after a reasonable number of iterations (<500). The procedure was used to compensate the triangular driving signal too. We added a sentence in the Supplementary note 4 to clarify these points.

4. Is it possible by employing the pulse shaping technic based on cross-phase modulation to impress an asymmetric step-like chirp ?

Authors: The answer is definitely positive; an asymmetric chirp would correspond to impressing two different absolute values of the slope over the negative/positive slope portions. However, providing that the net jump (of total amplitude $2u_0$) is left unchanged, this introduces only a net tilt of the dynamics in the T-Z plane (i.e., an average non-zero velocity) without qualitatively modifying the observed dynamics. This is the reason why, without loss of generality, we restrict the experiment to symmetric step-like chirps. We have reported a new supplementary Fig. 6 to show this (Supplementary note 3).

5. In the manuscript is mentioned that the experimental square pulse provides a constant power over 2ns duration and 25MHz repetition rate, and in the normalization of the NLSE, the parameter TO scale with power

and is not directly linked to the initial pulse duration in order to control the magnitude on the chirp. For the sake of clarity, it would be better to report the actual pulse duration of the onset square pulse used in simulations modelling the experimental results.

Authors: We thank the Referee for the suggestion on which we agree. We have reported all the parameters of the square pulses used in the numerics in “Methods”. Clearly, it would be meaningless to scale T_0 with the actual duration of the square pulses as the latter should be ideally infinite.

6. In the caption of Fig. S4, the authors reported red stars as the experimental results from Ref.[6], while in panel S4(d) the same results are referred to Ref. [8]. Furthermore, in the same panel, the blue cross results indicate different figures with respect to the Fig. 2 of the main text. I suggest checking it.

Authors: We thank the Referee for noticing. After renumbering the References as initially meant, the correct Ref. is [8]. We have also corrected the link of blue crosses to experimental figures of the main text.

7. Panels (a) and (b) of Fig. 2. Which is the input condition of the power density for these two examples of RW-c-DSW regime? Is it an asymmetric case with a stepwise profile?

Authors: Panels (a,b) of Fig. 3 of the revised manuscript (originally Fig. 2) illustrates the same type of dynamics, namely formation of a rarefaction-DSW pair. In the simplest case, this is induced by flat chirp ($u_0=0$) and a jump only in density/power (from 1 to $\rho_0 < 1$ in normalized units), which is by definition always asymmetric due to the fact that power (unlike chirp) is obviously positive. This is the case reported in panel b. However, the same dynamics can occur when a sufficiently small and symmetric jump in velocity/chirp accompanies the primary jump in power. This is indeed the case of panel a: indeed $u_0 = 0.5$ implies a finite symmetric stepwise chirp.

8. In the animation provided by the authors, y-and x-axis labels in the frequency chirp plot should be changed with u_0 and ρ_0 to be consistent with Fig. 2 in the main text of the manuscript and Fig. S4 in the supplementary.

Authors: We have changed the labels in the animation as suggested. We thank the Referee for the suggestion.

9. I suggest authors revise the reference list of their manuscript because the abbreviation form of some cited journals is not reported in the correct manner. For the sake of completeness, I would also suggest them to cite along with Ref. [34] even the following paper dealing with temporal Riemann pulse phenomena in nonlinear optical fibers:

S. Wabnitz, “Optical tsunamis: shoaling of shallow water rogue waves in nonlinear fibers with normal dispersion,” *J. Opt.* **15**, 064002(2013)

D. Bongiovanni, *et al.*, “Third-order Riemann pulses in optical fibers,” *Opt. Express* **28**(26), 39827-39840 (2020)

Authors: We have reviewed the reference list, in particular the abbreviations. We have also included few more references. Among these we added the fiber experiment by Bongiovanni et al. (Ref. [44]) pointed out by the Referee. As for the first theoretical reference suggested by the Referee, since third-order dispersion is not the focus of the paper, we have included several other theoretical and experimental contributions (now Refs. [33-39,57,63]), whose content is more relevant for the Riemann step-problem that we deal with in our paper (Riemann pulses and Riemann step-problem are indeed two different topics in spite of the fact that both are named after Riemann).

Summary of main changes in the revision:

1) We have split former Fig. 1 in two (Fig. 1 and Fig. 2) and adapted the text accordingly

2) We have inserted and/or expanded different paragraphs (red text in the revision) as suggested

3) We have expanded the bibliography inserting 10 additional relevant references

4) We have added a new subsection in Methods entitled “Numerical simulations”

5) We have expanded the Supplementary Material to address the strictly technical auxiliary points by adding a new subsection (S.M. Note 3, reporting additional simulations), and three new supplementary figures 2,6,7.

REVIEWER COMMENTS

Reviewer #1 (Remarks to the Author):

I have carefully read the revised version of the paper by Bendhamane and co-authors.

I appreciated the efforts of the authors to address my concerns.

While the introduction and the experimental section are now more accessible to a general audience, I still find that the theoretical description is quite difficult to follow. In particular figure 3 presents a large number of situations not investigated in the experiments making the reading not straightforward at all.

In my opinion this could penalize the paper, discouraging many potential readers. I strongly suggest the authors to further simplify this section, limiting the theoretical discussion in the main text to the situations which correspond to the experiment. The other cases should be moved in the Supplementary Material, with a great benefit for the readability of the manuscript.

Reviewer #2 (Remarks to the Author):

I thank the authors for carefully responding to all my queries. The manuscript is excellent and, in my view, deserves publication.

Reviewer #3 (Remarks to the Author):

This reviewer provided confidential remarks to the editor recommending publication of the manuscript.

Reviewer #1 (Remarks to the Author):

I have carefully read the revised version of the paper by Bendhamane and co-authors. I appreciated the efforts of the authors to address my concerns.

While the introduction and the experimental section are now more accessible to a general audience, I still find that the theoretical description is quite difficult to follow. In particular figure 3 presents a large number of situations not investigated in the experiments making the reading not straightforward at all.

In my opinion this could penalize the paper, discouraging many potential readers. I strongly suggest the authors to further simplify this section, limiting the theoretical discussion in the main text to the situations which correspond to the experiment. The other cases should be moved in the Supplementary Material, with a great benefit for the readability of the manuscript.

Authors: We thank the Referee for recognizing our efforts and to find that the introduction and the experimental section are more accessible. The Referee suggests to further simplify the theoretical section, limiting the discussion to the cases which are experimentally investigated. We find this comment pertinent, and we have modified the paper accordingly. We have removed two panels from Fig. 3, which illustrate the behavior of the dam-breaking case that was analyzed in previous works. We have also shortened the theoretical section, removing the description of the shock tube, dam-breaking problem.

Summary of main changes in the revision:

- 1) We have simplified Fig. 3 by removing 2 panels
- 2) We have shortened the theoretical section
- 3) In addition, we slightly changed Fig. 1, because the sketch of the RW was imprecise.

REVIEWERS' COMMENTS

Reviewer #1 (Remarks to the Author):

The authors have satisfactorily taken into account my remarks.

In my opinion, the present form of the manuscript is suitable for publication in Nature Communications.

Reviewer #1 (Remarks to the Author):

The authors have satisfactorily taken into account my remarks.

In my opinion, the present form of the manuscript is suitable for publication in Nature Communications.

We are happy to know that the Reviewer suggests the publication of the current version of the manuscript. We wish to thank this Reviewer for the constructive comments that helped us to improve the quality of our paper.